# State-led agricultural subsidies drive monoculture cultivar cashew expansion in northern Western Ghats, India

**Anushka Rege** ⓘ *◉, **Janice Ser Huay Lee**◉

Asian School of the Environment and Earth Observatory of Singapore, Nanyang Technological University, Singapore, Singapore

◉ These authors contributed equally to this work.
* REGE0002@e.ntu.edu.sg

## Abstract

Agricultural commodity production constitutes an important livelihood source for farmers but significantly contributes to tropical deforestation and biodiversity loss. While the socioecological effects of agricultural commodities such as palm oil, cocoa and coffee are well studied, the effects for commodities such as cashew (*Anacardium occidentale*) have received less attention. Global cultivated area for cashew increased rapidly from 526,250 ha in 1980 to ~5.9 million ha in 2018. India is the world's second largest cashew producer, with cashew farms often occurring adjacent to remnant forests. To mitigate deforestation for cashew expansion, it is necessary to understand present-day land use policies and management practices that drive this expansion. Through semi-structured interviews (n = 65) and a literature review on agricultural policies in India, we evaluated the role of state-led land use policies in cashew expansion and characterised present-day cashew farming systems in the Sawantwadi-Dodamarg landscape in India. Agricultural subsidies introduced from 1980s to 1990s encouraged cultivar cashew expansion and influenced land use conversion from rice and privately owned forest to cashew. Farmers preferred cultivar cashew as they produced higher yields faster, although they required more agrochemical inputs and were susceptible to pests and wildlife depredation. About 80% of farmers had planted cashew farms by clearing forests in the past 30 years and expressed interest to continue the same. Farmers avoided applying for government-sponsored compensation for crop losses due to wildlife depredation and chose instead to expand cultivar cashew into forested areas. Our study deepens the understanding of how government-led agricultural subsidies drive farmers' uptake of cashew cultivars, farmers' cashew management practices, and how these factors drive deforestation in this landscape at the state and farm level. We recommend further research with equitable stakeholder participation in cashew farming systems to devise sound planning for forest conservation and sustainability standards for the cashew industry.

**Data Availability Statement:** According to the NTU Institutional Board Review ethical guidelines, we had pledged complete confidentiality of respondents' identities and locations, given that our questionnaire included sensitive topics such as land ownership, human-wildlife interactions and

relationships between farmers and Government bodies such as the Forest Department. We have deposited the minimal data in an anonymised, de-identified format with the DR-NTU Dataverse, which can be accessed freely: https://doi.org/10.21979/N9/FZSWO4 This is also in accordance to Nanyang Technological University's guidelines on human subjects' (read:https://www.ntu.edu.sg/research/research-integrity-office/institutional-review-board/guidelines/data-repository ) which do not allow uploading of traceable data in either open- access or locked format, onto the NTU Data Repository, also known as DR-NTU (Data).

**Funding:** This work was supported by the Singapore Ministry of Education Academic Research Fund Tier 1 RG145/19 Grant secured by J.S.H.L. The funders had no role in study design, data collection and analysis, decision to publish, or preparation of the manuscript. This research was supported by the Earth Observatory of Singapore via its funding from the National Research Foundation Singapore and the Singapore Ministry of Education under the Research Centres of Excellence initiative. This work comprises EOS contribution number 415.

**Competing interests:** The authors have declared that no competing interests exist.

## Introduction

Agricultural production is necessary to fulfil nutritional requirements for human populations and global food security [1]. The agricultural sector also forms the economic backbone of many low-income countries in which millions of farmers are employed [2]. However, agriculture is also a major driver of deforestation and biodiversity loss across the tropics as cropland expansion increased since 1999 to present-day, with approximately 27% of global forest disturbance from 2001 to 2015 associated with the production of agriculture commodities [3, 4]. Managing agricultural production to achieve multiple objectives of providing food, supporting livelihoods and protecting biodiversity and forests is a pertinent challenge especially in tropical, low-income countries [5].

In recent years, the growth in global agricultural production is largely driven by commercial and export-oriented commodities such as meat, timber, palm oil, cereals, and soybeans [6]. The substantial role of governmental agricultural policies in influencing commodity expansion and their environmental and social outcomes on tropical land systems has been documented in Vietnam [7], Ethiopia [8], and Indonesia [9]. In particular, the provision of government subsidies to promote the expansion of the agricultural commodity sector has led to biodiversity and forest loss, over-consumption of land and water resources, thereby affecting the social fabric of communities that derive sustenance from lands affected by such subsidies [10, 11]. Globally the total value of agricultural subsidies harmful to biodiversity were estimated at USD 451 billion, out of which USD 100–230 billion were considered as most harmful to biodiversity [12].

While the socioecological drivers and outcomes in crop systems such as oil palm [13–15], coffee [16–18], and cacao [19] have been well studied, the effects of expansion of other agricultural crop systems in biodiversity hotspots are relatively less understood [3, 20]. 'Superfoods' such as coconuts, cacao, quinoa, açai, avocados, and almonds are in high demand across middle- and high-income countries that are far from regions of crop cultivation and involve significant environmental costs [20]. Such commodity crops have a constant international market due to year-round supply of commodities. One such agricultural commodity that has high global demand and is understudied for its socioecological effects is cashew (*Anacardium occidentale L.*). Native to the Brazilian *cerrado* [21], cashew is presently cultivated in 33 countries, including countries that are 'biodiversity hotspots' [22, 23]. Global land area for cashew cultivation between 1980 and 2018 has seen an eleven-fold increase from 526,250 ha to 5,972,724 ha [22]. As of 2019, Côte d'Ivoire, India, Tanzania, Benin and Indonesia had the largest land areas under cashew cultivation while USA, Germany, Netherlands, UK, Australia and Belgium were major importers [22, 24].

Cashew grows well in poor, nutrient-deficient soils and does not need intensive labour for most part of the year except during the harvest season [21]. Majority of cashew cultivation across the tropics is carried out in small-scale farms as opposed to large-scale, industrial plantations [25–30]. The two main products from cashew trees are cashew nuts and the cashew apple or pedicel [21]. Besides the cashew nut and pedicel which are sold for human consumption, cashew products are also used as animal feed and nut oil for industrial purposes [31–34].

India plays a major role in global cashew production since it has the second largest extent of cashew cultivation in the world (1,003,601 ha), preceded by Côte d'Ivoire and followed by Tanzania, Benin and Indonesia [22]. India is also an agrarian economy with ~86% of all farmers being smallholders who own less than 2 ha of land [35]. The states of Maharashtra, Goa in the west, and Andhra Pradesh and Orissa to the east are the main cashew producing states in India, accounting for ~60% (613,270 ha) of India's total land under cashew cultivation as well as cashew nut production (~449 million kg) in 2017 [36, 37].

The national average yield of cashew nuts in India is 900 kg/ha and is dependent on the type of cashew grown as well as climatic and soil conditions [38]. Two types of cashew crop are grown in India—the common type and cultivar type, the latter has numerous variants that have been selected for their high yielding properties [21]. The common type refers to the local, 'wild' cashew that has traditionally been grown. The cultivar type refers to the artificially propagated varieties, and they produce stable yields within three to four years as compared to seven years that the common cashew needs. However, they also require more agro-chemical inputs as compared to the common cashew which is more resilient to pests [21].

In the last decade, the extent of planted cashew in India has increased from 893,000 to 1,105,000 ha [22], with most expansion occurring within the states that constitute the Western Ghats, a globally recognized biodiversity hotspot [23]. The Western Ghats has lost 33,579 km$^2$ of forest cover between 1920s and 2013, with conversion to plantations as the main driver of deforestation, followed by agriculture and land degradation [39]. Questions related to how cashew expansion occurs, how these farming systems operate, and how cashew affect forest cover in the Western Ghats are poorly studied and require urgent attention given that monoculture cashew plantations are on the rise in these landscapes [40] and forests in this region have significant biodiversity value [41, 42].

wHere, we aim to review the historical processes of the expansion of cashew cultivation in the Sawantwadi- Dodamarg landscape (1,396 km$^2$) in Sindhudurg district, south Maharashtra. The forests within the Sawantwadi- Dodamarg landscape serve as a wildlife corridor for large mammals such as Bengal Tigers, Common Leopards and Asiatic Elephants but are largely unprotected and threatened as cashew monocultures increase. We present a historical review of cashew cultivation expansion with a focus on the role of state-led policies in driving this expansion and discuss an initial characterization of present-day cashew farming systems in India.

## Materials and methods

### Study site: Sawantwadi- Dodamarg, Sindhudurg, Maharashtra, India

Maharashtra is the top cashew producing state in India with a total of 191,450 ha under cultivation, corresponding to ~18% of India's cashew cultivated area [43, 44]. Sindhudurg district accounts for almost a third of the total land area under cashew cultivation in Maharashtra, with ~28% of all croplands in Sindhudurg district (total land area of Sindhudurg: 5,207 km$^2$) under cashew cultivation [45]. Sindhudurg district is divided administratively into eight *tehsils* (sub-districts), of which Sawantwadi (896 km$^2$) and Dodamarg (500 km$^2$) comprised our study site (Fig 1).

The district Sindhudurg, in which the two *tehsils* are located, is also a highly biodiverse region located within the Western Ghats-Sri Lanka global biodiversity hotspot [23]. In 2010, forest area in Sindhudurg was 503 km$^2$ [46]. This was reduced to 386 km$^2$ in 2013 [45] and 331 km$^2$ in 2018 according to a survey conducted by the Agriculture Department of Maharashtra [47]. Most forests in Sindhudurg are under communal or private land ownership and are not legally protected [41]. Reserved forests are forest patches that fall under the state Government protection, and activities such as hunting, firewood collection and grazing are banned.

Dodamarg *tehsil* overlaps with the Tillari landscape which is a large mammal corridor connecting three Protected Areas—Radhanagari Wildlife Sanctuary (in Maharashtra state), Mhadei Wildlife Sanctuary (in Goa state) and Bhimgad Wildlife Sanctuary (in Karnataka state) [41, 42]. The dominant forests here are moist deciduous with semi-evergreen vegetation along riparian areas.

The landscape in this region is hilly and undulating with elevations ranging from 0 to 1030 m above sea level [42, 48]. Annual rainfall ranges from 2300 mm to 4300 mm. The climate is

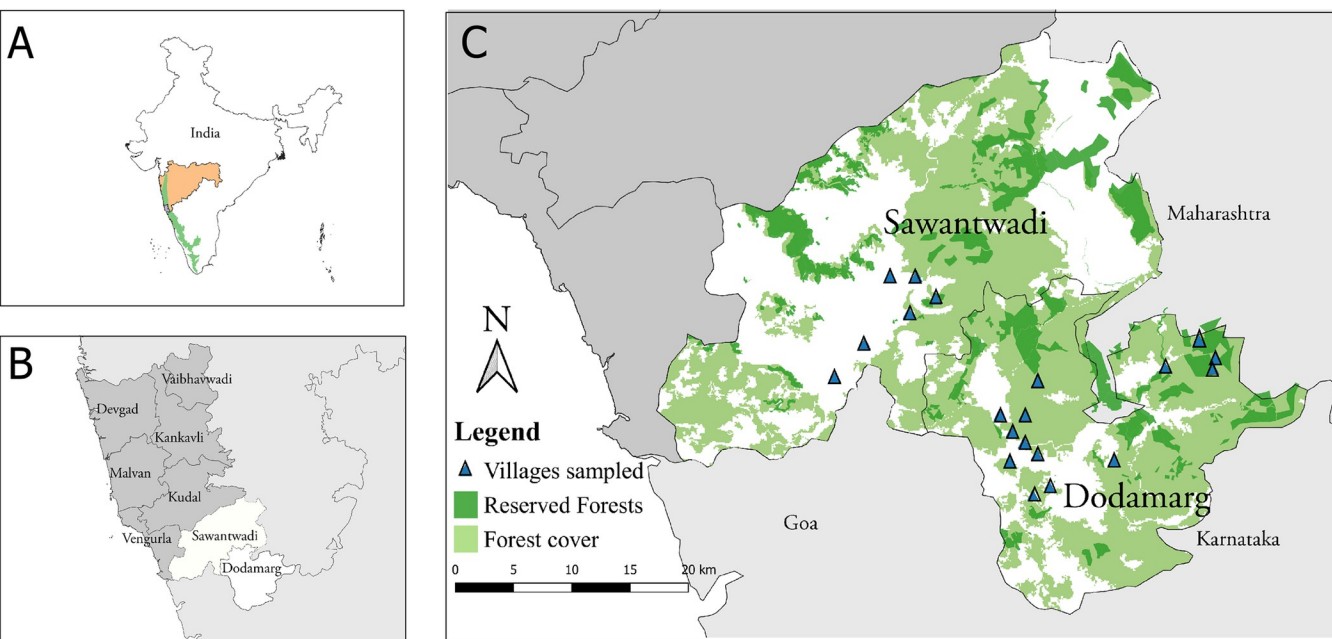

**Fig 1.** A map depicting (A) the state of Maharashtra (in orange) and the extent of the Western Ghats (in green), (B) all the *tehsils* within Sindhudurg district with the tehsils of Sawantwadi and Dodamarg in white, and (C) the locations of the villages where interviews were conducted (blue triangles), the extent of forest cover (in light green) and Reserved Forests (dark green) within Sawantwadi and Dodamarg (Data sources: https://bhuvan.nrsc.gov.in/home/index.php; https://gadm.org/data.html; Maharashtra Forest Department; https://daac.ornl.gov).

humid throughout the year, with 4 seasons: Winter (December to February), Summer (March to May), Monsoon (June to September), Post monsoon season, also known as retreating monsoon (October to November). The mean annual temperatures range from 16˚C to 35˚C [42].

The dominant ethnicity of the local population is Hindu Maratha (S1 Table, Census of India 2011). The literacy rates of Sawantwadi and Dodamarg are 86.71% and 75.37% respectively and the total proportion of population that lives in rural areas are 75.9% and 60%, respectively [49]. Agriculture is the primary occupation in this region and major crops include cashew, rubber, bamboo, and rice, with smallholdings of coconut, banana, and areca nut. In more recent years, pineapple is increasingly cultivated [42, 50]. The villagers depend on forests for firewood and minor forest produce such as bamboo, cane, honey, and medicinal plants [42].

### In situ interviews and bibliographic information

To develop our understanding of the historical processes and land use policies that influenced cashew expansion, we relied on in situ data collection and bibliographic information.

We conducted a preliminary field survey followed by semi-structured interviews between September and December 2018. Our preliminary field survey included site visits to cashew farms, cashew processing facilities where we interviewed cashew farmers and processors using an exploratory research framework. We asked questions related to the land use history of cashew farming systems, agricultural policies, farming practices and challenges faced by farmers in cashew farming systems. Information from our preliminary survey was used to develop a semi-structured interview to collect both qualitative and quantitative information on the socioeconomic background of cashew farming households and their farm management practices. More specific questions on farm management such as the size of the farm, weeding

frequency, use of chemical fertilisers and pesticides, use of common and cultivar cashew, fencing used to deter wildlife, land tenure and rights, market prices as well as the land use history of their cashew farms were included (S1 Appendix for full questionnaire). Some of our questions were open-ended and we followed up with our respondents on any new insights on cashew cultivation. We received ethics approval from the Nanyang Technological University Institutional Review Board (IRB reference number 2018-11-017).

We used snowball sampling [51] to facilitate the recruitment of cashew farmers for our interviews. We did not have a documented list of the cashew farmers in our study site, hence snowball sampling was preferred. The interviews were conducted during mornings or evenings and some interviews were conducted in the afternoon when farmers would return home or take a break from farming activities. Each interview lasted 30–40 minutes. We took notes on paper and did not use any means of digital recording as this made respondents uncomfortable during the interview. The interviews were conducted in the local Malvani dialect and Marathi language. The interviewing team comprised of one female and one male interviewer, and they could both speak Malvani and Marathi. Our interview notes were taken using English, Malvani and Marathi. The interviewing team did not include other locals from the study site to minimise any influence of familiarity on the respondents' answers. We interviewed a total of 65 cashew farmers, all from unique households (45 respondents from Dodamarg and 20 respondents from Sawantwadi) across 20 villages—13 in Dodamarg and 7 in Sawantwadi (S2 Table).

At the beginning of each interview, we introduced ourselves, the purpose of the study, informed the respondent that no monetary compensation will be provided and that the respondent is free to not answer any questions at any stage of the interview. We began the interview after we received oral consent from the respondent. Written consent would involve requesting for their signatures, which would make them suspicious of our motives; generally, locals in this region are averse to written consent or agreements. The ethics approval from the Nanyang Technological University Institutional Review Board had approved use of oral consent and this was documented in the dataset as well. None of the respondents we approached declined to be interviewed.

We used three approaches to build a historical overview of cashew expansion in the Sawantwadi-Dodamarg landscape from 1498 to present day and highlighted the role of government agricultural policies in the region's land use change. Firstly, we included insights received from our interviews with farmers (one of which was a former civil servant employed by the district's agricultural office, and one had a State-approved cashew nursery business as a secondary occupation). We also corresponded regarding the history of cashew introduction in India with a cashew-product businessman who is a cashew history chronicler. Secondly, we obtained cashew agriculture and production information, including open-access Government reports from websites of agencies such as the Maharashtra Agriculture Department, the Indian Council of Agricultural Research-Directorate of Cashew and Cocoa Development. Lastly, our review was informed by scientific papers and grey literature written in English (e.g., popular articles, reports, news articles and book chapters) which were drawn from Google and Google Scholar. We used Google and Google scholar since it also hosts useful grey literature which may not be found on other search engines. We used combinations of key search words such as "cultivar cashew", "dwarf cashew", "agricultural subsidies", "India", "Maharashtra", "tropics", and "cashew expansion".

## Data analysis

We used Excel and R software for quantitative analysis [52], and NVivo software [53] for transcribing and analysing qualitative data. Notes in Malvani and Marathi were translated into

English. Each interview was included as a 'case' and coded under 'nodes'. Each node refers to a categorisation of the information in the interview. For example, a statement such as '*the money I earn from selling my cashew yields is sufficient*' would be coded as '*fair prices received for cashew produce*'. Themes were further identified from the qualitative data which included: (i) Prevalent farm and private forest management practices: this pertains to the management practices that farmers employ to maintain privately owned forests and cultivate cashew in their farms. (ii) Cashew economy: this refers to monetary concerns that farmers may have, such as labour availability and market prices of cashew. (iii) Farmers' motivation for farming cashew: This refers to motivations and reasons respondents cited for growing or wanting to expand cashew farms. (iv) Human-wildlife interactions: this pertains to the nature of interactions that farmers experience with wildlife and how they perceive wildlife. These interactions could be neutral, positive, or negative from the perspective of the respondents. (v) Farmers' grievances: this refers to issues, obstacles, and persistent complaints that farmers may express regarding farming, land rights and human-wildlife interactions.

To characterise cashew farmers, we used the farmers classification system currently employed by the Indian Government [35]. This classification system categorises farmers based on their total land area owned: (i) marginal farmers who own less than 1 ha of land, (ii) small farmers who own 1–2 ha of land, (iii) semi-medium farmers who own 2–4 ha of land, (iv) medium farmers who own 4–10 ha of land, and (v) large farmers who own above 10 ha of land. The term 'total land area' includes all land owned, such as cultivated and uncultivated land. This classification system was used for analysis as it is used across India for the purpose of distributing subsidies, loans, benefits, and all decision-making with regards to resource-allocation by government agencies [35]. We categorised our respondents based on this classification system and compared the similarities and differences in cashew farming systems across these farmer categories.

## Results and discussion

### Overview of cashew expansion in South Maharashtra

Cashew was brought to India by the Portuguese who first arrived in 1498 to India, and in 1510 in Goa. It is believed that the Portuguese seafarers carried cashew with them from Brazil to other Portuguese colonies such as Mozambique in Africa in the sixteenth and seventeenth centuries [21]. It is debatable that Arab traders spread it during their sea travels across present-day cashew growing countries that also happened to be former Portuguese colonies (Vaz 2019, *pers comm*.). A popular belief is that cashew cultivation was promoted by Portuguese for soil erosion control, although this is refuted by modern historians [36]. Another plausible explanation for cashew cultivation in India is that the native communities, who faced unclear land tenure in the face of colonialism, could keep and live off land if they cleared virgin forest tracts for cultivation of hardy trees such as cashew (Vaz 2019, *pers comm*.). This is akin to present-day agricultural frontiers created via cashew expansion in Guinea-Bissau in order to establish land ownership [54]. In India, cashew spread across much of peninsular India and even to Northeastern states, notably Meghalaya. Cashew gained quick acceptance as a crop in India as it could be grown across soil types with low maintenance.

In South Maharashtra, common cashew was grown traditionally for centuries, but first gained large-scale commercial popularity across the world in the 1970s as it required less agricultural inputs compared to other crops such as rice and finger millet [21]. Farmers now grow rice and cashew, alongside crops such as coconut, arecanut, and bananas [42, 50]. Cultivar varieties of cashew were first developed at the onset of the Green Revolution in 1974 in India [55]. Cultivar cashew does not grow tall like common cashew, but branches horizontally,

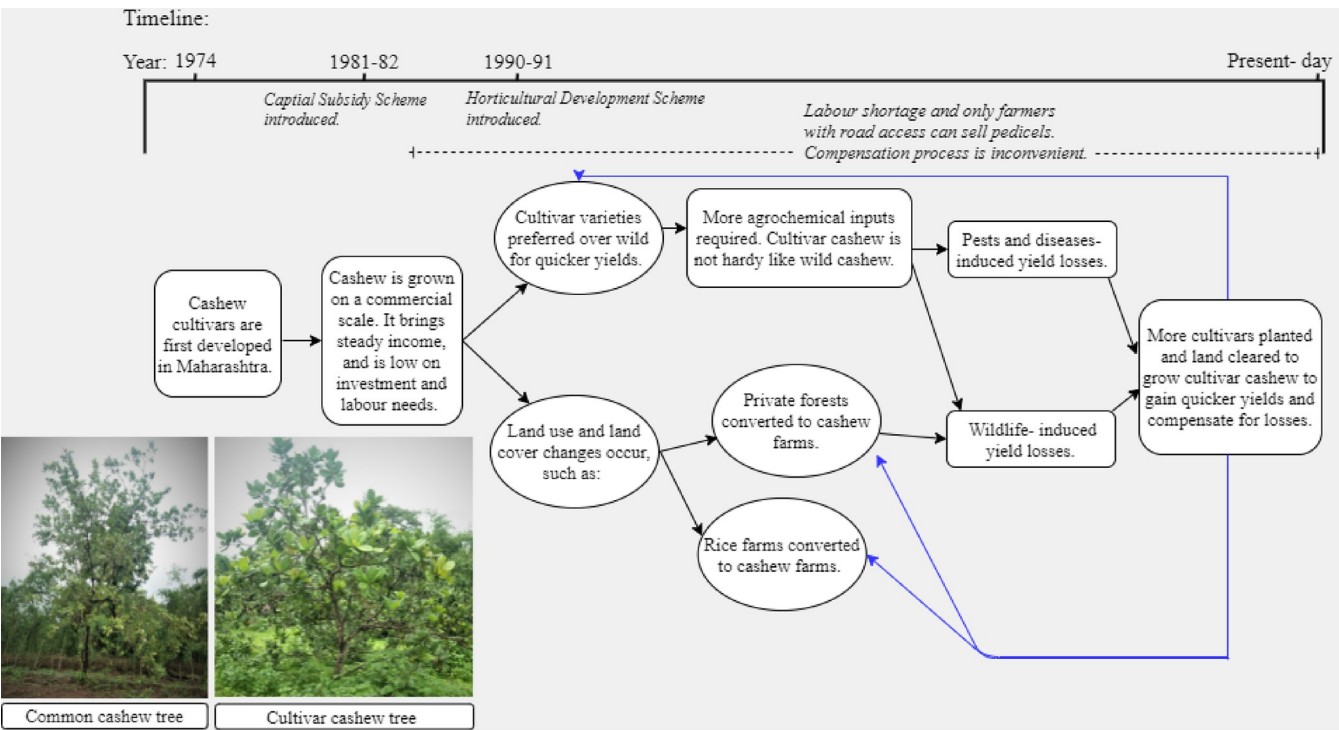

**Fig 2. A timeline of how agricultural subsidy policies led to increased adoption of cultivar cashew in South Maharashtra.** Cultivar cashew is favoured for quicker yields but is vulnerable to losses due to pests and wildlife. To compensate for these losses, more privately owned forests are cleared to grow cultivar cashew in the bid for quick yields and monetary profits. The blue arrows represent feedbacks in planting cultivar cashew in the pursuit of profits, which leads to even more losses.

hence it is also called 'dwarf cashew'. The main purpose of the cultivar varieties was to increase cashew yield in the shortest time and maximise yields from the available agricultural land. Soon after India's first cultivar cashew was introduced in 1974 in Maharashtra, other states such as Goa, Karnataka, Orissa, Tamil Nadu, Andhra Pradesh, Kerala and West Bengal developed their own cultivar cashew varieties [44]. Cultivar varieties are in demand and preferred by farmers as they fruit and flower within 3 to 4 years compared to 7 to 10 years required by common cashew, thus cultivators generate higher profits in shorter time [21].

Two state-led policies, both introduced by the Government of Maharashtra, played a significant role in the adoption and expansion of cultivar cashew in South Maharashtra: (1) the Capital Subsidy Scheme and (2) the Horticulture Development Scheme (Fig 2).

The Capital Subsidy Scheme was introduced in 1981–82 [56, 57] and aimed at improving the livelihoods of marginal farmers and economically weaker sections of the farming community by providing small loans for agricultural production. Capital subsidy was available from nationalised, co-operative and development banks in the form of loans for small and marginal farmers for horticultural crops, wherein 33% of costs incurred to cultivate cultivar cashew were reimbursed. This Capital Subsidy Scheme was available to farmers who had to meet several requirements including owning less than 1 ha of irrigated land or 2 ha of unirrigated land, earning from their cultivated lands net returns of at least 3,600 INR (47.91 USD equivalent), and must be small and marginal farmers, agricultural labourers, scheduled castes and scheduled tribes and differently abled individuals, as determined by the Government [58].

The Horticulture Development Scheme was introduced in 1990–91 and linked to the National Rural Employment Guarantee Act 1972 [58, 59]. It provided for both labour as well

as material costs (including machinery and agrochemicals) for a full or partial rate for farmers based on which farm classification they belonged to. Farmers from marginal and small holder categories, scheduled castes, scheduled tribes and nomadic tribes were eligible for full subsidy for material and labour costs, while farmers from the remaining categories were eligible for 75% subsidy for the material cost including equipment and tools needed for cashew farming, such as grasscutters, spray tank and nozzle, and a full subsidy for labour cost. This subsidy would be applicable for a minimum of 0.10 ha and a maximum of 4 ha [58].

Since the introduction of these two schemes, adoption of cashew cultivars has increased rapidly in Maharashtra. By 1990–91 alone, 31,883 horticultural cultivators had availed of the Capital Subsidy Scheme subsidy and by 1992, 47,961 ha of total land area was covered (by various horticultural crops of which cashew is one) under this scheme [58]. By 1996–97, 13,695 ha of total land area was growing cultivar cashew under the Horticulture Development Scheme [58]. In the state of Maharashtra, eight cultivar varieties—ranging from Vengurla-1 to Vengurla-8, were introduced from 1974 to 2001 by the Vengurla Regional Fruit Research Centre based in Vengurla, Maharashtra and affiliated to the Maharashtra State Government [50]. Of these, Vengurla-4 and Vengurla-7 were, and continue to be, most preferred in south Maharashtra, as they fruit within 4 years and yield 17.2 and 18.5 kg of cashew nuts respectively per tree as opposed to common cashew which takes 7 to 10 years to yield a stable production of about 10 to 12 kg [55]. The cultivar cashew saplings bought from the nurseries affiliated with the State Government are eligible for subsidy, while common cashew is not promoted as such.

While these cultivar varieties provide higher yields and mature faster, they are less resilient than the common cashew as they require pesticide and fertiliser inputs and are vulnerable to pests. In the Sawantwadi-Dodamarg region, cultivar varieties are widely used by cashew farmers who also reported cashew losses from pests and crop depredation by wildlife. Farmers in this region respond to these losses by planting more cultivar varieties in the hopes of recovering their losses and gaining profits (Fig 2). Growing cashew over privately owned forests to make up for crop losses is a short-sighted strategy as it does not reduce the chances of more loss and possibly increases crop depredation by wildlife.

Besides India, cultivar cashew varieties have been promoted in countries such as Brazil and Tanzania [26, 55, 60, 61]. In Nigeria and Tanzania, cultivar varieties are preferred for higher cashew nut weight, fetching high prices [60, 61]. The pattern of land use change in our study site can be contrasted with the case of cultivar cashew expansion in the Northeast Brazilian *caatinga* biome [26]. Although cashew prices reduced in Brazil, there was a surge in the production of cultivar cashew with pest-resistant qualities requiring less agrochemical inputs in the *caatinga* landscape; ensuring stable profits from smaller landholdings, ultimately leading to a reduction in deforestation rates [26].

In the Sawantwadi-Dodamarg region, a small proportion of forests are classified as 'Reserved Forests' by the Government and these are afforded some level of protection by the Forest Department. However, the majority of forest lands in this region fall under individual ownership. Although hunting of wildlife is not permitted by law irrespective of the land use in question (62), locals are legally entitled to convert their private forests into agricultural land. This remains a conservation challenge for the landscape as forests face land use conversion and are not legally protected.

The fact that cashew needs minimal care, the development of cultivar varieties that produce short-term profits, coupled with the state-led agricultural subsidies introduced in 1980s and 1990s have influenced farmers to convert privately owned forests to cultivar cashew plantations, leading to deforestation in the Sawantwadi-Dodamarg landscape (Fig 2).

It is worth noting that on a global scale, the 2011–2020 Conventional Biological Diversity (CBD) signatory countries (including India) have committed under Aichi Target 3 to ensure

| | Marginal (< 1 ha) n = 9 | Small (1-2 ha) n = 7 | Semi-medium (2-4 ha) n = 12 | Medium (4-10 ha) n = 17 | Large (≥ 10 ha) n = 20 |
|---|---|---|---|---|---|
| Cashew variety | Cultivar = 5 Common = 1 Combination = 3 | Cultivar = 1 Common = 1 Combination = 5 | Cultivar= 3 Common = 2 Combination = 5 | Cultivar = 5 Common = 1 Combination = 10 | Cultivar = 4 Common = 1 Combination = 15 |
| State subsidies availed | 3 | 2 | 4 | 11 | 14 |
| Agrochemical usage | Pesticides = 2 Fertilisers = 5 | Pesticides = 5 Fertilisers = 4 | Pesticides = 6 Fertilisers = 8 | Pesticides = 10 Fertilisers = 14 | Pesticides = 8 Fertilisers = 10 |
| Inherited farm ownership | 6 | 6 | 10 | 15 | 16 |
| Forest clearing | 8 | 6 | 8 | 16 | 14 |

**Fig 3. Typology of cashew farmers in Sawantwadi-Dodamarg landscape, South Maharashtra, India.** The table shows the number of respondents (in green boxes) within each farmer category based on attributes (as shown in blue boxes). The category 'cashew type' refers to the number of farmers growing common, cultivar or a combination of the two. Similarly, the category 'agrochemical usage' refers to the number of farmers using chemical pesticides and fertilizers in their farms. 'State subsidies', 'Inherited farm ownership' and 'Forest clearing' refer to the number of farmers that availed subsidies for the Government, inherited farmland from their family and cleared forest within the past 30 years for cashew cultivation, respectively. For detailed figures on all responses and no responses, please refer to S4 Table.

that all subsidies are free of practices that might harm biodiversity, a commitment that might continue further in future CBD frameworks [62]. The non-compliance to this commitment has been documented in many several instances worldwide, and our study adds to that list [62, 63].

## Present-day cashew farming systems in Sawantwadi-Dodamarg

Based on our semi-structured interviews in the Sawantwadi-Dodamarg landscape with cashew farmers (S3 Table for farmers' household characteristics), the majority of the respondents (75%, n = 49) were semi-medium, medium and large holders based on the Indian Government's farmer classification system (Fig 3). Some medium and large holders had high extents of uncultivated forested areas (ranging from 15% - 99% of their total land area for 29 out of 37 medium and large holder respondents) but comparatively smaller areas under cashew cultivation (Fig 4). These forests, being privately owned, face a potentially higher chance of being converted into monoculture cultivar cashew in the near future. The forests are owned privately, hence the farmers have legal rights to clear them as they see fit.

The size of land holdings under cashew cultivation in our study site are similar to across other regions such as Tanzania, Ghana, Guinea- Bissau [28–30], Brazil [26, 27] and Indonesia [25].

**Prevalent farm and private forest management practices.** The harvest season for cashew in the Sawantwadi-Dodamarg landscape occurs from January to May each year. Farmers planted new saplings in the monsoon months of June and July. Weeding is typically done one to three times around August, December, and/or January to April. However, the number of times farmers chose to weed their farms in a year was subject to availability of farming tools, labour and capital needed to employ labour.

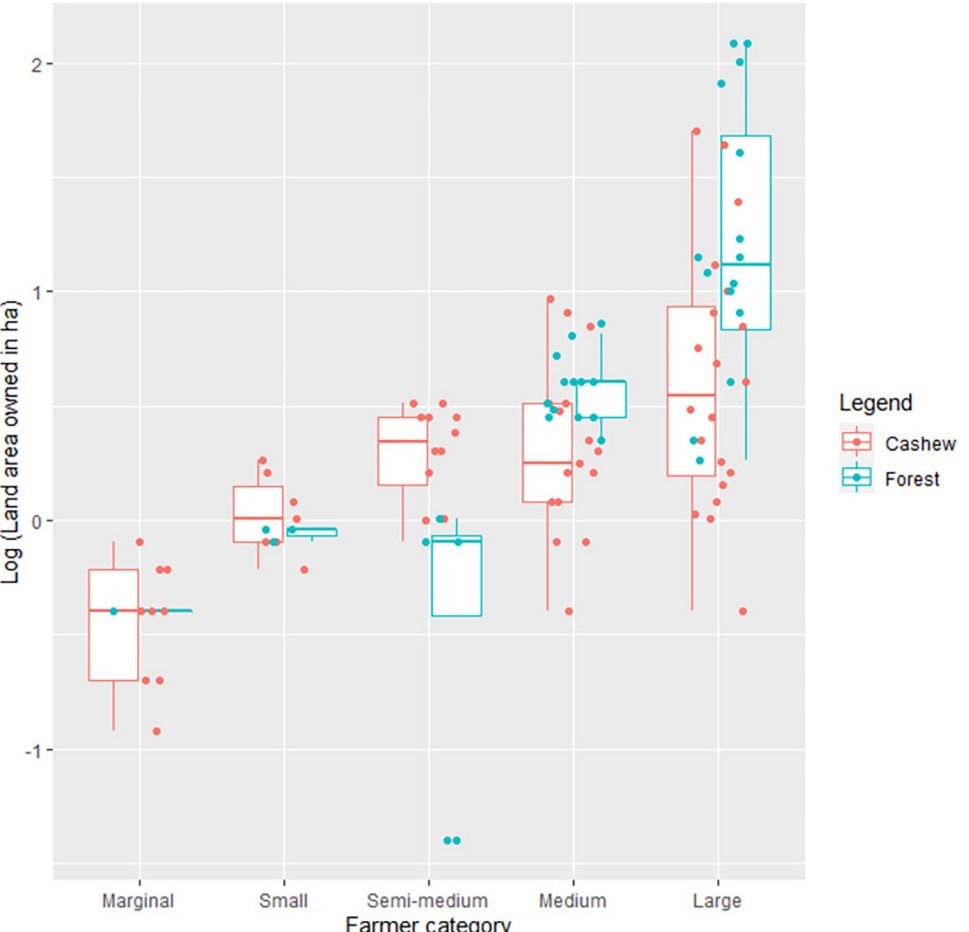

**Fig 4. Boxplots of land area under cashew cultivation and forests owned by all farmers' categories in the Sawantwadi-Dodamarg landscape, South Maharashtra, India.**

Half of the interviewed farmers (52%, n = 34) applied for and received government subsidies for growing cultivar cashew. All farmer groups preferred cultivar cashew over common cashew. Cultivar cashew in combination with common cashew were most planted among our respondents (58%, n = 38). Respondents stated that the number of cultivar individuals planted were much higher than that of the common cashew (for every 4 cultivar cashew individuals planted, 1 common cashew individual would be planted). Farmers who grew only cultivar varieties comprised 28% (n = 18) of our respondents and farmers who grew only common cashew comprised 9% (n = 6) of our respondents. The Vengurla-4 (V-4) and Vengurla-7 (V-7) cashew varieties were the most preferred across farmer categories, since V-4 and V-7 yielded high quantities within 4 years of planting. The common cashew on the other hand, takes 7 to 10 years to start flowering and fruiting stably enough to provide farmers with a stable income.

Over two-thirds of interviewed farmers (64%, n = 42) used chemical fertilizers and around half of farmers (46%, n = 30) used chemical pesticides. Chemical pesticides and fertilisers were favoured over organic ones because chemical options are easily available and there is little to no awareness regarding organic options. The key chemical ingredients in the products included endosulphan and phosphates, and the volume of pesticides and fertilisers used varied from 1 to 20 ml per 1 litre water solvent (1 litre used for 2 to 3 cashew individuals).

Across all groups, farming was done on mostly inherited land (86%, n = 56), i.e. passed down through families. Inheriting farmlands and privately owned forest lands gives farmers legal rights to clear the privately owned forests for cashew expansion, which would not be possible on leased lands.

The lack of awareness regarding organic farming and prolonged use of chemicals could adversely affect soil health in the long run, rendering the lands unfit for any cultivation. There is a need to also engage with farmers to understand how privately owned forests could contribute to their livelihoods, reducing deforestation and the reliance on cultivar cashew alone.

**Cashew economy.** In our study region, farmers either sell their cashew in local markets or access a cashew factory indirectly through a merchant. Merchants visit the villages during fruiting season (February to May). The major local markets in this region are the towns of Dodamarg, Banda, Sawantwadi; the major local cashew factories include the Desai factory at Kalane village, followed by Kesarkar and Kanekar factories in the town of Banda (Fig 5). The cities of Mumbai, Bengaluru, Pune, and Mapusa serve as important markets for the cashew nut produced from this region (Fig 5). Farmers transport their cashew nut produce either on motorcycle or bus, while merchants travel across villages and use vans and trucks for transportation.

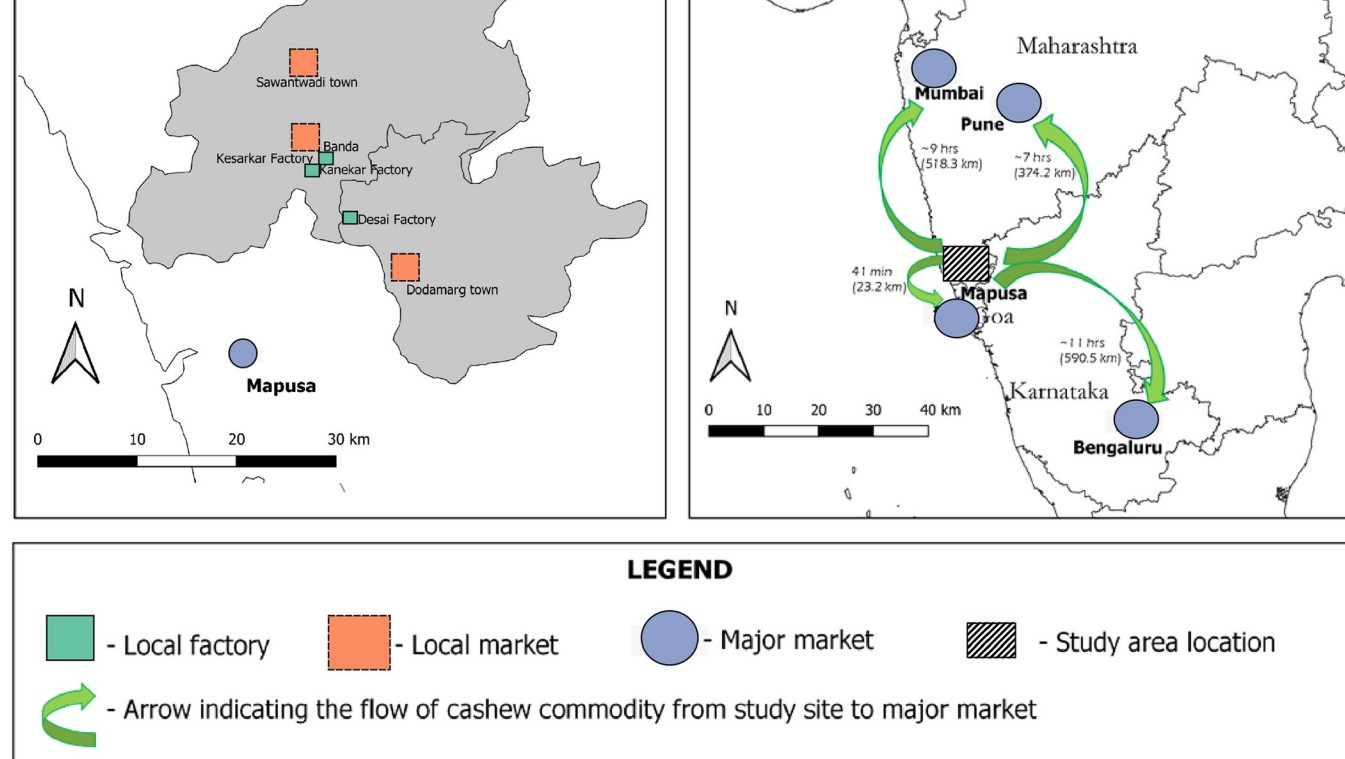

**Fig 5.** The figure illustrates the locations of (A) local cashew factories and markets in the Sawantwadi-Dodamarg landscape (in grey), and (B) major markets where the cashew commodity produced from the study site is transported to, along with the flow of the commodity in green arrows also showing the distance from study site to the major market and time taken to reach these markets by car. Distances and travel time to Mapusa and Bengaluru were estimated from Dodamarg town, while distances to Mumbai and Pune were estimated from Sawantwadi town (Data sources: https://bhuvan.nrsc.gov.in/home/index.php; https://gadm.org/data.html).

The average price of cashew received by farmers from merchants and local factories in the year of 2018 was 167.75 INR (SD = 14.44) or 2.36 USD (SD = 0.20) per kilogram (with prices ranging from 156.5 INR (SD = 24.48) to 170.65 INR (SD = 12.22) per kilogram). The price variation depends on factors such as pre-existing relations with the merchants and the quality of cashew nut produce. These rates were higher than the national average of 126 INR, as well as the average rates for other states (ranging from 102 to 142 INR) in India [37]. The average price per kilogram was also higher than the global average wholesale rate per kilogram of 1.42 USD in 2019 [64].

Over two thirds (63%, n = 41) of our farmers felt that the prices they received in payment for their cashew nut produce from the merchant/factory were fair while slightly over one fourth (22%, n = 14) felt that prices received were less in comparison to the time and money they invested. Five (8%) respondents felt that they were at the mercy of the market price fluctuations, and the rest of our respondents did not respond to this question on fair prices. 4 of the total 65 respondents (6%) reported transport problems such as infrequent public transport and high travel costs to bigger markets.

Only 23 (35%) respondents who had quick, easy access to road transport reported selling cashew pedicel to *feni* factories, which were transported by vans to the state of Goa for *feni* production. Fruits were sold at the low cost of an average of 1.17 (SD = 1.12) INR or 0.02 (SD = 0.02) USD) per kg (cashew fruits are sold in boxes, with fruit within each box weighing 12 kg). Farmers in interior villages, lacking easy and quick transport did not sell the fruit as it decays within a day post-harvest and needs to be transported soon after—in contrast to cashew nuts which can be stored for weeks before processing in factories. *Feni* can be brewed legally only in the state of Goa, hence farmers lacking easy road access are unable to easily sell the fruit produce.

Over two third of the farmers interviewed (66% or n = 43) employed local labour. Farmers who hired labourers from other Indian states were mostly semi-medium, medium, and large holders (23% or n = 15). Labourers tend to come from the states of Kerala, Jharkhand, Orissa, Bihar and Karnataka. During our interviews, farmers stated that labour shortage is an issue as more young adults were moving to urban centres for further education and employment. Family members and local fellow villagers contributed to each other's farm labour. There was an informal understanding amongst villagers that they would help one another due to labour shortage.

The proportion of farmers that were part of co-operatives was highest for large farmers (n = 7), followed by marginal (n = 3) farmers. Farmers' co-operatives are self-governed farmer groups with the motive to provide farmers a platform to share farming knowledge and experiences. They are also a reliable medium for raising and safeguarding emergency capital and providing assistance during a crisis such as crop losses or extreme weather events. Each co-operative may have farmers from several local villages. The State Government provides support to farmer co-operatives in the form of capital, subsidized loans, as well as trainings and workshops on farming practices. Farmers who joined co-operatives reported that while these groups may be able to supply loans, in practice they do not have a system in place to effectively disseminate information on good farming practices, policy and information related to subsidy, or to improve market access.

Presently the cashew industry in India has no uniform regulations and stipulated guidelines for sustainable production [21] in contrast to the coffee industry, which has started taking some steps towards sustainable production [65, 66]. Many Indian cashew nut companies adhere to FairTrade regulations–but none have pledged to any deforestation commitments [67]. Cashew cultivation, being a profitable venture, stands the risk of contributing to forest loss, particularly in regions where forests are privately owned and not legally protected.

**Farmers' land use decisions for farming cashew.** About 80% (n = 52) farmers had cashew farms that were planted over forests in the past 30 years and had expressed that they plan to continue to clear forests to plant cashew if they had sufficient funds to do so. Affluent farmers have the means to buy land from smaller farmers to expand land under cashew cultivation. Two of our respondents from the large land holder category stated that they owned large cashew monoculture farms of over 40 ha and had expanded these by buying smaller adjoining patches of farmed and forested lands from marginal and small farmers. On a global level, cashew is fast replacing native habitat such as forests in Benin [68], *caatinga* in Brazil [26] and savannah woodlands in Guinea-Bissau [69]; and crops such as cocoa in Ghana [29] and rice in Guinea- Bissau [70].

About 64% of our respondents reported owning private forests. The average forest land area owned was 0.04 ha (SD = 0.13), 0.38 ha (SD = 0.47), 0.22 ha (SD = 0.40), 3.07 ha (SD = 2.19) and 28.21 ha (SD = 41.77) across marginal, small, semi-medium, medium and large holders respectively. 40% (n = 26) of our respondents depended on their private forests for extraction of firewood. Many of them stated that the ongoing "*forest clearing*" and "*deforestation*" for cashew expansion meant that "*habitat is lost*" and this is bound to have negative effects in terms of crop depredation and human-wildlife conflict. Majority of respondents stated that they saw wild animals in agricultural lands and in their villages, and not just within forests, consistent with a camera-trapping study which showed that wildlife used cashew plantations as habitats [40]. However, farmers still planned on clearing private forests for cashew cultivation. A young respondent who recently took to cashew cultivation said that: "*We should have started growing cashew 10 years ago, but our eyes have opened now*".

Some medium and large holders have much larger uncultivated forested areas but comparatively smaller areas under cashew cultivation- such farmers could play a larger role in private forest conservation. These conservation challenges are similar in other tropical low-middle income countries (such as Mexico, Papua New Guinea, and Tanzania) where large extents of forests are owned by local communities, and including these communities in forest conservation is key [71].

**Human-wildlife interactions.** Farmers reported that cultivar cashew varieties are less sturdy compared to common cashew and experienced more frequent crop depredation by wildlife. Majority of our farmers (98% or n = 64) stated that they suffered crop losses. Apart from wildlife, farmers (24% or n = 16) also mentioned insect pests such as Stem borer (*Plocaederus ferrugineus*) and Tea mosquito bug (*Helopeltis* spp.) as the main reason for crop losses. Farmers (17% or n = 11) from Sawantwadi also stated that losses due to pests had an equal if not a greater role in cashew crop losses.

Across all respondents, the most reported wildlife for crop depredation were sambar, gaur, porcupine, macaques and langurs and wild pig respectively (Fig 6). The nature of crop loss varied, from loss of cashew nuts to bark damage and gradual uprooting of trees/saplings. The terms for macaques and langurs ("*vanar*" and "*keldi*") were used interchangeably due to the species having similar appearances. Civets, rodents and bats were reported on a broad level and could not be identified to the exact species level. Most of the farmers used a combination of mitigation measures to deal with wild animals in their farms, which ranged from scaring the wildlife with scarecrows and firecrackers, installing various types of fences, to installing lighting in their farms. Farmers mostly used wooden fencing (20%) to protect their farms, although other alternatives such as bamboo (15%), solar powered (18%) fences and wired (12%) fences were also utilized. 39 (60%) respondents reported visiting their farms every day. 23 (40%) respondents reported that they visited farms only occasionally (from a couple of times a week to rarely; they would regularly visit their farms only in the fruiting season in the month of May, to collect the produce), while 3 respondents did not respond.

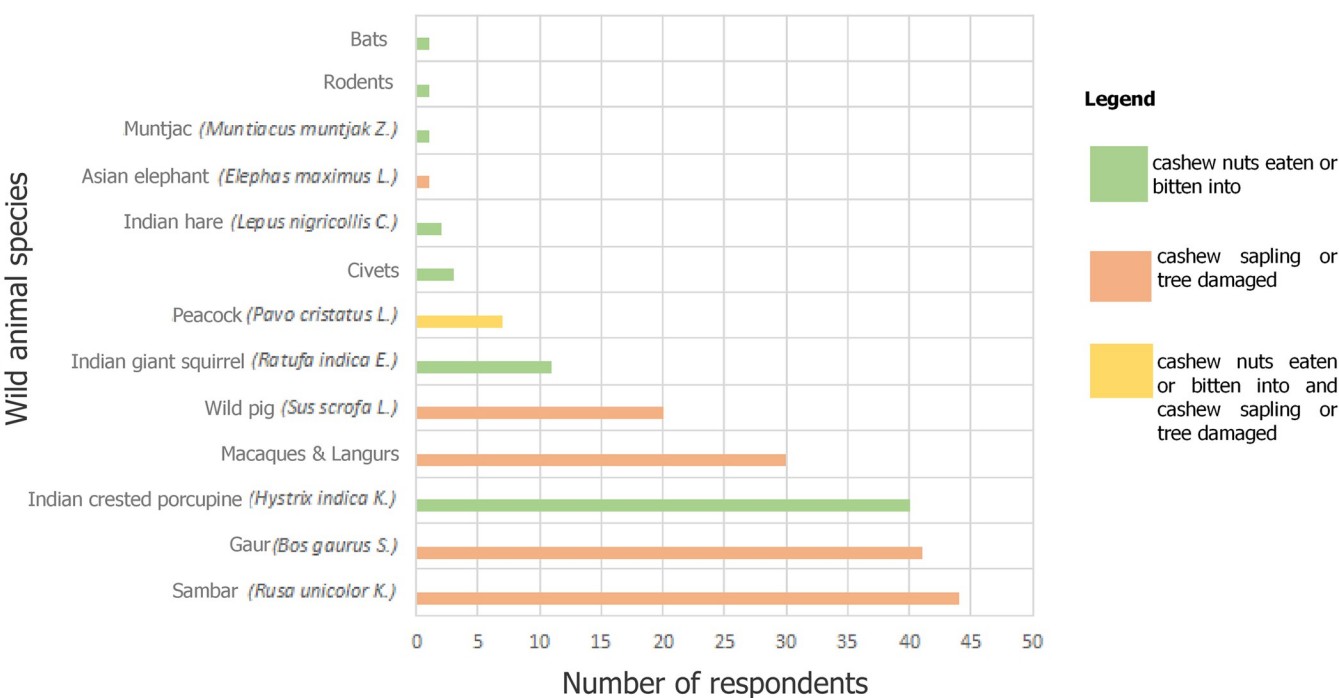

**Fig 6. Wildlife reported to cause cashew crop loss and the nature of damage reported across 65 respondents.**

The sapling stage of cashew plants is crucial since that is when they are most prone to damage by wildlife. There is a need to delve deeper into the issue of human-wildlife interactions to evaluate which conflict mitigation strategies might work best to protect young cashew saplings in the Sawantwadi- Dodamarg landscape.

**Farmers' grievances.** Farmers who experienced crop depredation by wildlife did not file for Government-sponsored compensation as the process of filing for compensation was "*tiresome*", "*complicated*" and "*lengthy*", and entailed proving land ownership for the farm in which crop depredation occurred, which was another tedious bureaucratic process. Studies on compensation schemes elsewhere in the Western Ghats found that the farmers held similar views [72].

Farmers did not keep exact records of how much losses they suffered due to crop depredation Some farmers reported that the compensation received was not worth the total losses incurred and travel costs to visit the Forest Department (FD) office. Majority of farmers (78%, n = 51) stated that the negative interactions with wildlife [73] have gotten worse compared to five to ten years ago and that they would like the issue of crop depredation to be better addressed by the Forest Department. Farmers were aware that this human-wildlife conflict was a result of rising deforestation (30%, n = 20) while some attributed higher wildlife populations (as compared to five to ten years ago) to bans on hunting and clearing of Reserved Forests. Farmers expressed that they would like the FD to translocate animals from farms into forests and set aside land for wildlife. They stated that educational campaigns conducted by the Agricultural and Forest Departments would be of help in land management and human-wildlife conflict mitigation. However, studies have shown that translocation has not been proven as a safe measure to reduce negative human-wildlife interactions and might even exacerbate conflict [74].

Another reason they avoid filing for compensation is that doing so requires them to produce a legal document to prove land ownership of the said farm in question, along with

signatures of all owners of that land requesting for compensation. Farmers often co-inherit land with other siblings who may reside in other villages or cities, making this legal requirement for all land-owner signatories inconvenient. Farmers stated the need for reforms in the current systems of land ownership and tenure. Ensuring a faster compensation process which requires only the current 'keeper' of the land, besides enabling regular proactive dialogue among stakeholders on how to manage and mitigate human-wildlife interactions would be useful steps in this direction. Insurance schemes for crop losses could serve useful as well [72, 74].

Other grievances which farmers had were related to a perceived lack of training facilities and workshops on the best farming practices conducted by the State Agricultural Department and local authorities. The Indian cashew market has not yet experienced the demands of organic cashew from the consumer-end the way that coffee market has, for instance. The idea of growing 'organic' cashew has only recently begun to be of interest to farmers, albeit on a very small scale.

## Conclusions

Cashew cultivation is in high global demand and supports livelihoods and the economy in the Sawantwadi-Dodamarg landscape. Our results demonstrate that the majority of the cashew farmers in the Sawantwadi-Dodamarg landscape farm on inherited land, prefer to grow cultivar cashew and use chemical pesticides and fertilisers in their farms.

The present-day land use system of the Sawantwadi-Dodamarg landscape has been shaped by the historical land use policies for cultivar cashew expansion. We could expect similar patterns of cashew expansion in other cashew-growing Indian states, where cultivar varieties were similarly promoted for producing quicker yields. The conversion of forests and rice land uses to cashew in our study site is similar to the land use changes seen in west African countries, the most notable one being that of the present-day conversion of forests to cashew in Guinea-Bissau [29, 69, 70]. We recommend stakeholder discussions among the Forest Department, Agricultural Departments and bodies and cashew farmers be conducted to consider reviving wild cashew in farms to reduce reliance on cultivar varieties, explore insurance schemes for crops and conduct more studies on human-wildlife interactions to mitigate any conflict events, and to consider long-term engagement with locals to find alternative livelihoods to conserve privately owned forests through responsible eco-tourism.

Since the cashew industry is in its infancy in developing sustainable production practices and contributes to the loss of native habitat, further research that investigates the processes governing cashew expansion in the tropics, the socioeconomic conditions of cashew farming communities, the effects of cashew expansion on biodiversity and the potential to set up a cashew sustainability initiative would be valuable. Research along these directions would greatly inform further policy and management and would be key to informing inclusive conservation action [75]. Furthermore, with the recent increase in adherence to deforestation-free commodity production, there is scope for the cashew industry to evaluate and modify its current practices to ensure compliance of adequate socioenvironmental standards, from the bottom-up farmers till the very consumer-end [76].

Our study demonstrates the role of government-led agricultural subsidies in driving farmers' uptake of cashew cultivars and provides an initial characterization of present-day cashew farming systems in India. We demonstrate how land use policies at the state level and cashew management practices at the farm level interact to drive deforestation in the landscape of Sawantwadi-Dodamarg in the state of Maharashtra, India. Forest conservation in the Sawantwadi-Dodamarg landscape presents a unique challenge since the forests are owned by people,

who have strong motivations to expand monoculture cultivar cashew farms to increase their incomes and compensate for crop losses from wildlife and pests. Understanding how state-led land use policies drive local-level decision-making on forest conversion for agriculture as well as developing a preliminary characterization of these farming systems are crucial first steps to begin formulating engagement plans with local landholders for conservation action in this landscape.

## Supporting information

**S1 Table. An overview of Sawantwadi and Dodamarg *tehsils* (Data sources: 10th Agricultural Census, 2015–16; Census of India, 2011).**
(DOCX)

**S2 Table. Number of respondents interviewed per village.**
(DOCX)

**S3 Table. Characterization of family and farm attributes across Small-, Medium-, and Large-holders.**
(DOCX)

**S4 Table. Detailed typology of cashew farmers (n = 65) in the Sawantwadi-Dodamarg landscape, South Maharashtra, India.**
(DOCX)

**S1 Appendix. The survey design which was used to interview cashew farmers in Sawantwadi and Dodamarg.**
(DOCX)

## Acknowledgments

We are grateful to N. Desai, P. Desai and S. Desai and their families for logistic support. A.R. thanks her mother for logistic support during the beginning of fieldwork. Thanks to N.T.L. Lim and D.A. Wardle for useful suggestions during data collection and analysis, A. Jayadevan and S. Bodhankar-Warnekar for useful inputs for making figures, V. Sadekar for sharing photos of cashew trees, and M. Patil, V. Patil, H. Vaz and N. Kulkarni for sharing useful contacts and insights. We thank M. Gangal, Y. Fatimah, S.W. Smith, L. Frias, K.W. Yuen, S.A Ismail and K. Deshpande for useful suggestions and comments on the manuscript.

## Author Contributions

**Conceptualization:** Anushka Rege, Janice Ser Huay Lee.

**Data curation:** Anushka Rege, Janice Ser Huay Lee.

**Formal analysis:** Anushka Rege, Janice Ser Huay Lee.

**Funding acquisition:** Janice Ser Huay Lee.

**Investigation:** Anushka Rege.

**Methodology:** Anushka Rege, Janice Ser Huay Lee.

**Project administration:** Anushka Rege, Janice Ser Huay Lee.

**Resources:** Anushka Rege, Janice Ser Huay Lee.

**Supervision:** Janice Ser Huay Lee.

**Validation:** Anushka Rege, Janice Ser Huay Lee.

**Visualization:** Anushka Rege, Janice Ser Huay Lee.

**Writing – original draft:** Anushka Rege.

**Writing – review & editing:** Anushka Rege, Janice Ser Huay Lee.

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
