## [Decision Letter · Decision Letter 0]

26 Jan 2022

PONE-D-21-34207State-led agricultural subsidies drive monoculture cultivar cashew expansion in northern Western Ghats, IndiaPLOS ONE

Dear Dr. Rege,

Thank you for submitting your manuscript to PLOS ONE. After careful consideration, we feel that it has merit but does not fully meet PLOS ONE’s publication criteria as it currently stands. Therefore, we invite you to submit a revised version of the manuscript that addresses the points raised during the review process.

ACADEMIC EDITOR: Please carefully address all the comments of the reviewer

We look forward to receiving your revised manuscript.

Kind regards,

Arun Jyoti Nath

Academic Editor

PLOS ONE

Journal Requirements:

2. We note that Figure 1 and 5 in your submission contain [map/satellite] images which may be copyrighted. All PLOS content is published under the Creative Commons Attribution License (CC BY 4.0), which means that the manuscript, images, and Supporting Information files will be freely available online, and any third party is permitted to access, download, copy, distribute, and use these materials in any way, even commercially, with proper attribution. For these reasons, we cannot publish previously copyrighted maps or satellite images created using proprietary data, such as Google software (Google Maps, Street View, and Earth). For more information, see our copyright guidelines: http://journals.plos.org/plosone/s/licenses-and-copyright.

1. You may seek permission from the original copyright holder of Figure 1 and 5 to publish the content specifically under the CC BY 4.0 license.  

Reviewers' comments:

Reviewer's Responses to Questions

**Comments to the Author**

1. Is the manuscript technically sound, and do the data support the conclusions?

Reviewer #1: Partly

2. Has the statistical analysis been performed appropriately and rigorously? 

Reviewer #1: N/A

3. Have the authors made all data underlying the findings in their manuscript fully available?

Reviewer #1: Yes

4. Is the manuscript presented in an intelligible fashion and written in standard English?

Reviewer #1: Yes

5. Review Comments to the Author

Reviewer #1: Comments to the Author

1. Is the manuscript technically sound, and do the data support the conclusions?

Discussion and conclusions could be improved.

2. Has the statistical analysis been performed appropriately and rigorously?

No significant statistical analysis was performed.

3. Have the authors made all data underlying the findings in their manuscript fully available?

In Availability Statement reference is made to the possibility of data request.

4. Is the manuscript presented in an intelligible fashion and written in standard English?

The English seems correct (though I’m not a native speaker). However, the text could be shortened allowing a faster reading.

Additional comments

The manuscript “State-led agricultural subsidies drive monoculture cultivar cashew expansion in northern Western Ghats, India” deal with the effects of agricultural policies in the cashew expansion and the consequences of these policies to forest conversion. This is a very interesting subject and an up-to-date problem, not only in this Asian region, but also in countries from other continents where cash crops acquired, in last decades, a preponderant role in countries’ economies. The manuscript presents interesting and very useful data that allows to evaluate some of the impacts of these public policies.

The manuscript has potential to be published, but, in the present version is too much descriptive and misses a more in deep analysis of the gathered information.

So, I suggest some changes that, to my point of view, can improve and valorise the work:

i) a more scientific and concise writing will be welcome (avoiding repetitions, superfluous details, that doesn´t make significant contribution to the paper, or even long sentences that can be replaced by less wordy sentences without losing information). Less can be more!

ii) improve the quality and the format of the tables. Moreover, tables, per si, are not very informative to the reader, so, other form of representation, such as graphics, could synthesize the most relevant information and would be more appellative.

Also, some of the information in the supplementary tables could be included in the main text.

ii) The merge of the results and discussion can be useful, but, to each point, it should have some discussion (e.g., the point ‘Human-wildlife interactions’ (line 487) do not have any discussion). Also, the overall discussion must be improved with a more in deep analysis and include some proposals / suggestions to overcome or mitigate the effects that state policies are having in forest ecosystems.

Besides these main shortfalls, I refer, below, other minor points:

Ln. 65 Anacardium occidantale – replace by Anacardium occidentale. In the first time it is referred in the main text (here) the authority should be included: Anacardium occidentale L.

Ln. 66 – ‘Brazilian cerrados’ – ‘Brazilian cerrado’; as far as I know, the plural is not commonly used.

Ln. 93 - Western Ghats, a globally recognized biodiversity hotspot (37). - Here you should cite 23. Myers et al. instead of reference 37.

Ln. 101 - (12)(10,11)(7)(8)(9)(43)(44,45) – are these references correctly placed here?

Ln. 122 - ‘Reserved Forests’ – as far I understand this involves a particular type of management that exists in India. Perhaps explain the meaning.

Ln. 134 – ‘vegetation along riparian areas’ - end point is missing

Ln. 141 – ‘The literacy rates of Sawantwadi and Dodamarg are 80% and 55% respectively’ - In S1 Table, it is referred: 79.4 and 75.37%. Please check the values in the tables and in the text to correct possible divergences.

Also in this table: ‘Population 1,47,466’ – do you mean 147,466?

Ln. 162 (S2 Appendix for full questionnaire) – it seems that some information gathered in field surveys is not referred in the manuscript. The data was insufficient? Not relevant? Explain or otherwise remove from the questionnaire. E.g. 43. How dense is the plot vegetation?

Ln. 241 (36) - end point is missing

Ln. 347 - sometimes is not clear the relation between the text and the mentioned figures. For instance, in ln. 347, ‘Some large holders had high extents of uncultivated forested areas (ranging from 15% - 99% of their total land area for 16 out of 20 large holder respondents) but comparatively smaller areas under cashew cultivation (Figs 3 and 4).’ The mention to Fig 3 seems unnecessary and fig. 4 doesn’t clearly illustrate the text. Where is the forested areas represented?

In S3 Table (Number of respondents interviewed per village), why use letters (A to T) instead the names of the villages?

S5 Table. – two rows are repeated at the beginning.

S6 Table – ‘Rodent spp. and Bat spp.’ - this is not correct. 'spp.' is used when the genus but not the species are known. In this case the genus is not referred. So, use only Rodent and Bat. Also, the authorities of species names should be included in the other referred species.

6. PLOS authors have the option to publish the peer review history of their article (what does this mean?). If published, this will include your full peer review and any attached files.

Reviewer #1: No

---

## [Author Response · Author response to Decision Letter 0]

8 Apr 2022

Journal Requirements:

Response: Thanks for this comment. We have reformatted the manuscript and all other submissions attached to meet the journal requirements. 

2. We note that Figure 1 and 5 in your submission contain [map/satellite] images which may be copyrighted. All PLOS content is published under the Creative Commons Attribution License (CC BY 4.0), which means that the manuscript, images, and Supporting Information files will be freely available online, and any third party is permitted to access, download, copy, distribute, and use these materials in any way, even commercially, with proper attribution. For these reasons, we cannot publish previously copyrighted maps or satellite images created using proprietary data, such as Google software (Google Maps, Street View, and Earth). For more information, see our copyright guidelines: http://journals.plos.org/plosone/s/licenses-and-copyright.

1. You may seek permission from the original copyright holder of Figure 1 and 5 to publish the content specifically under the CC BY 4.0 license. 

Natural Earth (public domain): http://www.naturalearthdata.com/’

Response: Thanks for raising this issue. We used DIVA (https://www.diva-gis.org/Data) freely downloadable and usable shapefiles for figures 1 and 5. We realise this issue may have occurred due to the preprint available online (https://osf.io/preprints/socarxiv/p6cge/) with the license ‘CC-BY Attribution-NonCommercial-ShareAlike 4.0 International’. We have deleted the figures 1 and 5 from the preprint. Please let us know if this issue still persists and what more we could do to address the same. 

Specifically, we used GADM shapefiles which are freely available for academic and other non-commercial purposes (see: https://gadm.org/license.html#:~:text=The%20data%20are%20freely%20available,academic%20research%20articles%20is%20allowed ). 

Response: This has been done, thanks for the comment. 

Response: Thank you for this suggestion. We have added a justification in our cover letter for sharing the anonymised data upon request: 

‘According to the NTU Institutional Board Review ethical guidelines, we had pledged complete confidentiality of respondents’ identities and locations, given that our questionnaire included sensitive topics such as land ownership, human-wildlife interactions and relationships between farmers and Government bodies such as the Forest Department. We have deposited the minimal data in an anonymised, de-identified format with the DR-NTU Dataverse, which can be accessed freely: https://doi.org/10.21979/N9/FZSWO4

This is also in accordance to Nanyang Technological University’s guidelines on human subjects’ (read:https://www.ntu.edu.sg/research/research-integrity-office/institutional-review-board/guidelines/data-repository ) which do not allow uploading of traceable data in either open- access or locked format, onto the NTU Data Repository, also known as DR-NTU (Data).’

 

Additional comments:

The manuscript “State-led agricultural subsidies drive monoculture cultivar cashew expansion in northern Western Ghats, India” deal with the effects of agricultural policies in the cashew expansion and the consequences of these policies to forest conversion. This is a very interesting subject and an up-to-date problem, not only in this Asian region, but also in countries from other continents where cash crops acquired, in last decades, a preponderant role in countries’ economies. The manuscript presents interesting and very useful data that allows to evaluate some of the impacts of these public policies.

Response: We thank the reviewer for their kind comments and the apt summary of our work. 

The manuscript has potential to be published, but, in the present version is too much descriptive and misses a more in deep analysis of the gathered information. So, I suggest some changes that, to my point of view, can improve and valorise the work:

5. a more scientific and concise writing will be welcome (avoiding repetitions, superfluous details, that doesn´t make significant contribution to the paper, or even long sentences that can be replaced by less wordy sentences without losing information). Less can be more!

Response: We thank the reviewer for pointing this out. We omitted the following sentences:

Lines 244- 246: Jan Huygen van Linchoten, a Dutch trader, documented that in 1584 people consumed 'cashew nuts' with coconut alcohol (57). Feni (cashew liquor) finds mention as early as 1583 in the Dutch spy’s Jacques de Coutre’s handwritten journals (58).

Lines 333- 336: Should crop depredation occur, locals are entitled to seek compensation from the Forest Department (67). However, most farmers prefer to convert privately owned forests to cultivar cashew farms rather than seek compensation, as they feel that the latter process is cumbersome (68). 

Lines 340- 344: While the farmers plant cultivar cashew for improving their socioeconomic status, cultivar cashew is not resilient and suffers from manifold losses, driving farmers into a cycle of expanding land under cultivar cashew, rather than planting the resilient common cashew or seeking compensation from the Forest Department for crop depredation losses (Figure 2).

Lines 359- 361: Since cashew crop brings in stable income and cultivar cashew is promoted via State sponsored subsidies, many farmers have cleared their forests to expand land under cultivar cashew, a trend that might continue in the future.

Lines 492-494: Privately owned forests in this region do not have any legal protection status but serve as important habitat and a corridor for wildlife movement (40,41). The support of cashew farmers in this landscape is crucial for any conservation action targeting privately owned forests.

Line 562: Cashew is a commercially important crop and in high demand globally (21).

We additionally modified some sentences: 

Lines 137-140: Dodamarg tehsil overlaps with the Tillari landscape which is a large mammal corridor connecting three Protected Areas - Radhanagari Wildlife Sanctuary (in Maharashtra state), Mhadei Wildlife Sanctuary (in Goa state) and Bhimgad Wildlife Sanctuary (in Karnataka state) (41,42).

Lin 146: The mean annual temperatures range from 16°C to 35°C (42).

Lines 210-211: We used Excel and R software for quantitative analysis (55), and NVivo software (56) for transcribing and analysing qualitative data.

‘Figure’ abbreviated to ‘Fig’ in Lines 314-316: Farmers in this region respond to these losses by planting more cultivar varieties in the hopes of recovering their losses and gaining profits (Fig 2).

Lines 319- 320: Besides India, cultivar cashew varieties have been promoted in countries such as Brazil and Tanzania (26,60,65,66).

Lines 337-340 modified to add reference to Figure 2: The fact that cashew needs minimal care, the development of cultivar varieties that produce short-term profits, coupled with the state-led agricultural subsidies introduced in 1980s and 1990s have influenced farmers to convert privately owned forests to cultivar cashew plantations, leading to deforestation in the Sawantwadi-Dodamarg landscape (Fig 2).

Lines 351-354 modified to mention Fig 3 and correct the grammar: Based on our semi-structured interviews in the Sawantwadi-Dodamarg landscape with cashew farmers (S4 Table for farmers’ household characteristics), the majority of the respondents (75%, n=49) were semi-medium, medium and large holders based on the Indian Government’s farmer classification system (Fig 3).

‘Figure’ abbreviated to ‘Fig’ in Lines 413-414: The cities of Mumbai, Bengaluru, Pune, and Mapusa serve as important markets for the cashew nut produced from this region (Fig 5). 

Lines 439-441 modified as: Only 23 (35 %) respondents who had quick, easy access to road transport reported selling cashew pedicel to feni factories, which were transported by vans to the state of Goa for feni production.

Lines 562-564 modified as: Cashew cultivation is in high global demand and supports livelihoods and the economy in the Sawantwadi-Dodamarg landscape.

Added a space between the sentence and the reference in Lines 570-572: The conversion of forests and rice land uses to cashew in our study site is similar to the land use changes seen in west African countries, the most notable one being that of the present-day conversion of forests to cashew in Guinea-Bissau (29,75,76).

Lines 596- 599 modified as: Understanding how state-led land use policies drive local-level decision-making on forest conversion for agriculture as well as developing a preliminary characterization of these farming systems are crucial first steps to begin formulating engagement plans with local landholders for conservation action in this landscape.

6. improve the quality and the format of the tables. Moreover, tables, per se, are not very informative to the reader, so, other form of representation, such as graphics, could synthesize the most relevant information and would be more appellative.

Also, some of the information in the supplementary tables could be included in the main text.

Response: We thank the reviewer for this suggestion. We have improved the quality and format of all the tables in the Supplementary Materials. 

Additionally, we also improved the formatting of Figure 3 in our manuscript as follows: 

Lines 363- 370: Fig 3. Typology of cashew farmers in Sawantwadi-Dodamarg landscape, South Maharashtra, India. The table shows the number of respondents (in green boxes) within each farmer category based on attributes (as shown in blue boxes). The category ‘cashew type’ refers to the number of farmers growing common, cultivar or a combination of the two. Similarly, the category 'agrochemical usage’ refers to the number of farmers using chemical pesticides and fertilizers in their farms. ‘State subsidies’, ‘Inherited farm ownership’ and ‘Forest clearing’ refer to the number of farmers that availed subsidies for the Government, inherited farmland from their family and cleared forest within the past 30 years for cashew cultivation, respectively. For detailed figures on all responses and no responses, please refer to S5 Table.

The reviewer suggested that some of the information in the supplementary tables could be included in the main text. We agree and hence we modified S6 Table into a bar chart and included it in the main text as Figure 6. Figure 6 now reads as follows: 

Line 522: Fig 6. Wildlife reported to cause cashew crop loss and the nature of damage reported across 65 respondents. 

Likewise, a mention of Fig 6 was added in lines 507-508: Across all respondents, the most reported wildlife for crop depredation were sambar, gaur, porcupine, macaques and langurs and wild pig respectively (Fig 6).

7. The merge of the results and discussion can be useful, but, to each point, it should have some discussion (e.g., the point ‘Human-wildlife interactions’ (line 487) do not have any discussion). Also, the overall discussion must be improved with a more in deep analysis and include some proposals / suggestions to overcome or mitigate the effects that state policies are having in forest ecosystems.

Response: We thank the reviewer for this suggestion. We have added a discussion paragraph at the end of each sub-section under ‘Results and Discussion’. 

Lines 404- 407: The lack of awareness regarding organic farming and prolonged use of chemicals could adversely affect soil health in the long run, rendering the lands unfit for any cultivation. There is a need to also engage with farmers to understand how privately owned forests could contribute to their livelihoods, reducing deforestation and the reliance on cultivar cashew alone. 

Lines 524- 527: The sapling stage of cashew plants is crucial since that is when they are most prone to damage by wildlife. There is a need to delve deeper into the issue of human-wildlife interactions to evaluate which conflict mitigation strategies might work best to protect young cashew saplings in the Sawantwadi- Dodamarg landscape. 

As the reviewer suggested, we also added our proposals / suggestions to overcome or mitigate the effects that state policies are having in forest ecosystems in lines 572-578: We recommend stakeholder discussions among the Forest Department, Agricultural Departments and bodies and cashew farmers be conducted to consider reviving wild cashew in farms to reduce reliance on cultivar varieties, explore insurance schemes for crops and conduct more studies on human-wildlife interactions to mitigate any conflict events, and to consider long-term engagement with locals to find alternative livelihoods to conserve privately owned forests through responsible eco-tourism.

Besides these main shortfalls, I refer, below, other minor points:

8. Ln. 65 Anacardium occidantale – replace by Anacardium occidentale. In the first time it is referred in the main text (here) the authority should be included: Anacardium occidentale L.

Response: Thanks for this correction, it has been made. 

Lines 69-70: One such agricultural commodity that has high global demand and is understudied for its socioecological effects is cashew (Anacardium occidentale L.).

9. Ln. 66 – ‘Brazilian cerrados’ – ‘Brazilian cerrado’; as far as I know, the plural is not commonly used.

Response: Thanks for this correction, it has been made.

Lines 70-72: Native to the Brazilian cerrado (21), cashew is presently cultivated in 33 countries, including countries that are ‘biodiversity hotspots’ (22,23).

10. Ln. 93 - Western Ghats, a globally recognized biodiversity hotspot (37). - Here you should cite 23. Myers et al. instead of reference 37.

Response: Thanks for this correction, we added the Myers et al. 2000 reference.

Lines 97-99: In the last decade, the extent of planted cashew in India has increased from 893,000 to 1,105,000 ha (22), with most expansion occurring within the states that constitute the Western Ghats, a globally recognized biodiversity hotspot (23).

11. Ln. 101 - (12)(10,11)(7)(8)(9)(43)(44,45) – are these references correctly placed here?

Response: Thanks for this correction, these references have been removed in line 106. 

12. Ln. 122 - ‘Reserved Forests’ – as far I understand this involves a particular type of management that exists in India. Perhaps explain the meaning.

Response: Lines 135-136 now read as: ‘Reserved forests are forest patches that fall under the state Government protection, and activities such as hunting, firewood collection and grazing are banned.’

13. Ln. 134 – ‘vegetation along riparian areas’ - end point is missing

Response: Thanks for this correction, we added an end point.

Line 140-141: The dominant forests here are moist deciduous with semi-evergreen vegetation along riparian areas.

14. Ln. 141 – ‘The literacy rates of Sawantwadi and Dodamarg are 80% and 55% respectively’ - In S1 Table, it is referred: 79.4 and 75.37%. Please check the values in the tables and in the text to correct possible divergences.

Also in this table: ‘Population 1,47,466’ – do you mean 147,466?

Response: Lines 148-149 read as: ‘The literacy rates of Sawantwadi and Dodamarg are 86.71% and 75.37% respectively and the total proportion of population that lives in rural areas are 75.9% and 60%, respectively (52).’ The S1 Table reads as follows:

S1 Table. An overview of Sawantwadi and Dodamarg tehsils (Data Sources: 10th Agricultural Census, 2015-16; Census of India, 2011)

Characteristics Units Sawantwadi Dodamarg

General information 

Area in km2 895.89 500.1

Number of villages - 82 62

Population number of individuals 147,466 48,904

Literacy rate % of total population 86.71 75.37

Population in rural areas % of total population 75.9 60

Dominant ethnicity % of total population Hindu (92 %) Hindu (96.21 %)

Number of households - 35,958 12,035

Farmers' information 

Marginal farmers (less than 1 ha) % of total farmer population 77.31 62.74

Small farmers (1.00 to 2.00 ha) % of total farmer population 10.91 13.69

Other farmers (above 2.00 ha) % of total farmer population 11.78 23.57

Total land area owned by marginal farmers % of total land area owned by all farmers 21.4 8.56

Total land area owned by small farmers % of total land area owned by all farmers 14.81 9.51

Total land area owned by other farmers % of total land area owned by all farmers 63.79 81.92

We have made corrections in the table. We thank the reviewer for the suggestions. 

15. Ln. 162 (S2 Appendix for full questionnaire) – it seems that some information gathered in field surveys is not referred in the manuscript. The data was insufficient? Not relevant? Explain or otherwise remove from the questionnaire. E.g. 43. How dense is the plot vegetation?

Response: The data were sufficient, but not fully relevant to the focus of the paper. Our questionnaire was exhaustive and detailed. We presented the original questionnaire as was used, to adhere to norms of transparency in reporting. We felt that adding too many details in the manuscript would make the paper unnecessarily long and deviate from the main point, which was to understand how state-led policies influenced land use conversion from forests to cashew monoculture plantations. The data shall be available freely in an anonymised format upon request, so that those readers interested in the details could easily obtain the data. 

16. Ln. 241 (36) - end point is missing.

Response: Thanks for this correction, we added an end point. Lines 246-248 read as: A popular belief is that cashew cultivation was promoted by Portuguese for soil erosion control, although this is refuted by modern historians (36). 

17. Ln. 347 - sometimes is not clear the relation between the text and the mentioned figures. For instance, in ln. 347, ‘Some large holders had high extents of uncultivated forested areas (ranging from 15% - 99% of their total land area for 16 out of 20 large holder respondents) but comparatively smaller areas under cashew cultivation (Figs 3 and 4).’ The mention to Fig 3 seems unnecessary and fig. 4 doesn’t clearly illustrate the text. Where is the forested areas represented?

Response: Thanks, we agree that the mention to Figure 3 is unnecessary and have omitted the mention of the same. Figure 4 has now been updated to include areas owned under the categories ‘Cashew’ and ‘Forest’, rather than total land owned and land owned under cashew. We hope this figure now aptly represents forested areas owned by cashew farmers. 

Line 354 -356 now reads as: ‘Some medium and large holders had high extents of uncultivated forested areas (ranging from 15% - 99% of their total land area for 29 out of 37 medium and large holder respondents) but comparatively smaller areas under cashew cultivation (Fig 4).’

The Fig 4 and caption have been modified as follows: 

Lines 372-373: Fig 4. Boxplots of land area under cashew cultivation and forests owned by all farmers’ categories in the Sawantwadi-Dodamarg landscape, South Maharashtra, India.

18. In S3 Table (Number of respondents interviewed per village), why use letters (A to T) instead the names of the villages?

Response: We used letters A to T instead of actual village names to conceal the village identities. We pledged to conceal respondent identities and locations in this study (in adherence to Institutional Review Board guidelines) and concealing village names is an important part of doing the same.

19. S5 Table. – two rows are repeated at the beginning.

Response: Thanks for this correction, we omitted the repeated rows. 

20. S6 Table – ‘Rodent spp. and Bat spp.’ - this is not correct. 'spp.' is used when the genus but not the species are known. In this case the genus is not referred. So, use only Rodent and Bat. Also, the authorities of species names should be included in the other referred species.

Response: Thanks for this correction, it has been made. We have included the authorities of species names in Figure 6 in the main text.

---

## [Decision Letter · Decision Letter 1]

16 May 2022

State-led agricultural subsidies drive monoculture cultivar cashew expansion in northern Western Ghats, India

PONE-D-21-34207R1

Dear Dr. Rege,

We’re pleased to inform you that your manuscript has been judged scientifically suitable for publication and will be formally accepted for publication once it meets all outstanding technical requirements.

Kind regards,

Arun Jyoti Nath

Academic Editor

PLOS ONE

Additional Editor Comments (optional):

Reviewers' comments:

Reviewer's Responses to Questions

**Comments to the Author**

1. If the authors have adequately addressed your comments raised in a previous round of review and you feel that this manuscript is now acceptable for publication, you may indicate that here to bypass the “Comments to the Author” section, enter your conflict of interest statement in the “Confidential to Editor” section, and submit your "Accept" recommendation.

Reviewer #1: All comments have been addressed

2. Is the manuscript technically sound, and do the data support the conclusions?

Reviewer #1: Yes

3. Has the statistical analysis been performed appropriately and rigorously? 

Reviewer #1: Yes

4. Have the authors made all data underlying the findings in their manuscript fully available?

Reviewer #1: Yes

5. Is the manuscript presented in an intelligible fashion and written in standard English?

Reviewer #1: Yes

6. Review Comments to the Author

Reviewer #1: Many thanks to the authors for the corrections and improvements made to the manuscript. I believe they contributed significantly to the readability and understanding of the article. Good work.

I only detected a small error on figure 3: the values of n and the sums of the categories 'cashew type' do not coincide regarding the typology of cashew farmers 'semi-medium' and 'medium'.

7. PLOS authors have the option to publish the peer review history of their article (what does this mean?). If published, this will include your full peer review and any attached files.

Reviewer #1: No

---

## [Editor Report · Acceptance letter]

19 May 2022

PONE-D-21-34207R1 

State-led agricultural subsidies drive monoculture cultivar cashew expansion in northern Western Ghats, India 

Dear Dr. Rege:

I'm pleased to inform you that your manuscript has been deemed suitable for publication in PLOS ONE. Congratulations! Your manuscript is now with our production department. 

Kind regards, 

on behalf of

Dr. Arun Jyoti Nath 

Academic Editor

PLOS ONE